# The Association of Minimally Invasive Surgical Approaches and Mortality in Patients with Malignant Pleuropericarditis—A 10 Year Retrospective Observational Study

**DOI:** 10.3390/medicina58060718

**Published:** 2022-05-27

**Authors:** Claudiu-Eduard Nistor, Camelia Stanciu Găvan, Alexandra-Andreea Ciritel, Alexandra Floriana Nemes, Adrian Ciuche

**Affiliations:** 1Department of Thoracic Surgery, Central Military Emergency University Hospital Bucharest, University of Medicine and Pharmacy “Carol Davila”, 010825 Bucharest, Romania; ncd5879@gmail.com (C.-E.N.); adrianciuche@gmail.com (A.C.); 2Family and Fertility Department, ESRC Centre for Population Change, Building 58, Room 2001, Faculty of Social Sciences, University of Southampton, Southampton SO17 1TW, UK; 3FMR Global Health, 22 Rue Jean Mermoz, 75008 Paris, France; 4Department of Neonatology, Louis Ţurcanu Clinical Emergency Hospital for Children, 300011 Timişoara, Romania; dr.alexandranemes@gmail.com

**Keywords:** malignant pleuropericarditis, minimal invasive surgery, survival analysis

## Abstract

*Background and Objectives:* Malignant neoplasms are common causes of acute pleuropericardial effusion. Pleuropericarditis denotes poor patient prognosis, is associated with shortened average survival time, and represents a surgical emergency. *Materials and Methods:* We analyzed the impact of two minimally invasive surgical approaches, the type of cancer, and other clinical variables on the mortality of 338 patients with pleuropericarditis admitted to an emergency hospital in Romania between 2009 and 2020. All patients underwent minimally invasive surgeries to prevent the recurrence of the disease and to increase their life expectancy. Log-rank tests were used to check for survival probability differences by surgical approach. We also applied univariate and multivariate Cox proportional hazard models to assess the effect of each covariate. *Results:* No significant differences were found in the 2-year overall survival rate between patients who underwent the two types of surgery. The multivariate Cox proportional regression model adjusted for relevant covariates showed that age, having lung cancer, and a diagnosis of pericarditis and right pleural effusion increased the mortality risk. The surgical approach was not associated with mortality in these patients. *Conclusion:* These findings open up avenues for future research to advance the understanding of survival among patients with pleuropericarditis.

## 1. Introduction

Malignant pleuropericarditis represents pleural and pericardial fluid effusions with a neoplastic etiology [1]. The majority of malignant pericarditides develop as complications in the evolution of malignant neoplasms. Various malignant neoplasms associated with pericardial and pleural effusions have been observed such as pulmonary, lymphomas, leukemias, malignant melanomas, breast, ovarian, prostatic, colonic, gastric, renal, and urinary bladder cancers [2,3]. However, primary cardiac or pericardial cancers associated with malignant pleuropericarditis are rare [2]. The prevalence of malignant pleuropericarditis varies between 12% and 23% of neoplastic patients [4], and this type of pericarditis accounts for 5–10% of all pericarditides [3].

The normal pericardial sac contains 10–50 mL of pericardial fluid acting as a lubricant between the pericardial layers [5]. The presence of more than 100–200 mL of pericardial fluid may cause a life-threatening pericardial constriction [5,6]. Pericardial effusions contribute greatly to the mortality of cancer patients because they are markers of spread, and treatment options are limited. Therefore, in cancer patients with pericardial effusions, emergency decompression is needed. [7]. Depending on the pleural effusion, pleuropericardic effusions can be small, moderate, or large, with imminent cardiac tamponade or constriction [1]. It is well known that the treatment is aggressive, but the prognosis for patients with malignant pericardial effusion continues to be poor and it is dictated by the primary entity/affliction [6].

The “gold standard” treatment for malignant pericardial effusion is still left undefined [1]. Approaches such as pericardiocentesis, a subxiphoid pericardial window, and the relatively recent minimally invasive thoracoscopic pericardial window, which drains the fluid, have been suggested as treatments [8,9]. Some papers examined survival in patients with either pericardial [10] or pleural effusion by surgical treatment type [11,12,13], but none of them diagnosed and treated surgically both effusions at the same time. To the best of our knowledge, this study is the first attempt to treat malignant pleuropericarditis with a minimally invasive thoracoscopic pleuropericardial window and a subxiphoid pleuropericardial window through mediastinoscopy surgery, and to compare the survival of these patients by the two types of surgery approaches.

## 2. Materials and Methods

### 2.1. Ethics Approval

Informed consent was obtained from all patients and data were anonymized prior to analyses. This study was approved by the Ethics Committee at the Central Military Emergency Hospital Dr. Carol Davila, no. 390/09.06.2020.

### 2.2. Study Population and Data Source

This retrospective study was conducted at the Central Military Emergency Hospital Dr. Carol Davila Bucharest, and the data were collected from July 2009 to July 2020. The study investigated time-to-death among 338 patients diagnosed with malignant pleuropericarditis aged between 19 and 94 years old. Most of these patients had been admitted to the Intensive Care Unit (ICU) for severe dyspnea, tachycardia, hypotension, paradoxical pulse, chest pain, fatigue, edema of the lower limbs, and arrhythmias. Demographic and clinical data were gathered using patients’ clinical records. Inclusion criteria were as follows: imaging confirmation of pleuropericarditis, cytological/pathology report confirming neoplasia, surgical treatment, age ≥ 18 years. Exclusion Criteria were: certified malignancy with isolated pericardial effusion, certificated malignancy with isolated pleural effusion, lack of cytology/pathology report confirming neoplasia, lack of surgical treatment, age < 18 years.

Data on the date-of-death were retrieved via operating protocols and from relatives via phone calls.

### 2.3. Therapeutic Management

The diagnosis of malignant pleuropericarditis was made using imaging, cytology, and/or pleuropericardial biopsy. Since all patients showed pericardial and pleural effusion, they underwent surgical treatment according to the pericardial and pleural guidelines [1,14]. Some patients were examined by computed tomography while others did not tolerate this examination. For them, we conducted cardiac and pleural echographic examinations, which established both the surgical indication and approach. As a result of all these examinations, a thoracoscopic pleuropericardial window and a subxiphoid pleuropericardial window through mediastinoscopy surgery were performed. Although some patients had had a previous diagnosis of cancer, pleural and pericardial biopsies were taken from all of them. Each patient underwent personalized diagnosis and therapeutic management.

There are no specific guidelines and recommendations in deciding when to use one type of surgery over another in managing malignant pleuropericarditis [10]. However, at our center, the subxiphoid minimally invasive approach is employed for patients with hypodiastolia and for those who had pleuropericarditis recurrence after thoracoscopic pleuropericardial window. For patients with hypodiastolia, orotracheal intubation was difficult or impossible to perform; moreover, orotracheal intubation did not allow for the creation of a pleural chamber. Therefore, the subxiphoid approach was preferred for hypodiastolia patients for easier access to the anterior face of the pericardium. This approach was chosen also because the supine position reduces the pressure created by the pericardial and pleural effusion on the heart and the lung parenchyma. As it was difficult for the effusion to drain through pericardiocentesis, patients with hypodiastolia underwent subxiphoid pericardial window with the insertion of an intrapericardial drainage tube, under local anesthesia.

We performed a thoracoscopic pleuropericardial window under general anesthesia for patients who did not show acute cardiac and respiratory failure (these patients did not have hypodiastolia). We performed thoracoscopic window surgery in patients who underwent selective intubation, and who supported the pressure of the pleural and pericardial effusion in the position of lateral decubitus. The lateral decubitus position of these patients sustained the pressure of the pleural and pericardial effusion, without determining the appearance of the pathophysiological phenomena of major compression on the mediastinum and of the contralateral lung. For patients with pericardial tamponade, we performed pericardiocentesis just before the surgery to drain the pericardium.

Chemical pleurodesis was performed for all patients to reduce the chances of malignant pericarditis recurrence [15]. Since betadine and talc are among the most used chemical agents for chemical pleurodesis [16,17,18], we used betadine in most patients and talc for those allergic to betadine. After surgical management of pleuropericarditis, all patients performed oncological treatment according to the personalized molecular diagnosis.

### 2.4. Measurements

This study includes patients’ characteristics and diagnosis types: age, sex, imaging diagnosis, clinical diagnosis, surgery type, comorbidities, presence of pericardial tamponade, and hypodiastolia. Patients’ cancer types include malignancies and metastases in sites other than pleura/pericardium. This paper also records pleurodesis treatment and pleuropericardities recurrence status after surgery. Survival is defined as the time from the date of diagnosis of pleuropericardial effusion to date-of-death. Imaging diagnosis distinguishes between echocardiography and computed tomography. The clinical diagnosis variable records patients with pericarditis and right pleural effusion, pericarditis and left pleural effusion, and pericarditis and bilateral pleural effusion. The surgery type variable distinguishes between a thoracoscopic pleuropericardial window and a subxiphoid window through mediastinoscopy. Patients’ comorbidity variables record the following categories: no comorbidity, hypertension, diabetes, atrial fibrillation, renal insufficiency, hepatic insufficiency, and chronic obstructive bronchopneumpathy. The presence of pericardial tamponade and of hypodiastolia are indicated by two dummy variables. We consider 12 groups of malignancies: lung cancer, sarcoma, esophageal cancer, lymphoma, pleural mesothelioma, ovarian, cervical, gastric, renal, and colorectal cancers. In the multivariate analysis, patients with lymphoma are grouped with those with leukemia because the latter is the only group with censored data by the end of the study observation (i.e., patients with leukemia were alive by the end of the study). Due to the small number of observations, patients with esophageal cancer are grouped with those with gastric cancer into one category, “esophageal/gastric cancer”. For the same reason, patients with sarcoma, colorectal, and renal cancer are grouped into one category, “other types of cancers”. Patients’ metastases other than pleura/pericardium indicate no other metastases, heart, liver, peritoneum, bone, upper kidney, or kidney. Pleurodesis treatment distinguishes between using talc and betadine. The pleuropericarditis recurrence variable distinguishes between those who had or did not have recurrence after surgery.

### 2.5. Outcome and Follow-Up Period

The main outcome of this study was mortality during the observational period 2 years from the date of diagnosis of pleuropericardial effusion. This study assessed the time-to-death from the date of diagnosis expressed in months.

### 2.6. Statistical Analysis

The patients contribute person-months to the data until they experience death or are being censored (e.g., either because they are lost, or alive by the end of the 48-month observation window). Whenever the date-of-death was in the same month as the date of diagnosis, we let these patients contribute with one person-month in calculating their survival probability.

Categorical variables are presented as absolute (n) and relative percentages (%). Age is the only numeric variable and is presented as the mean and its standard deviation (SD). All variables were compared by the two types of surgery, using the Chi-square test for categorical variables and Student’s t-test for continuous variables.

We analyzed the mortality risk differences across surgery types using Kaplan–Meier plots, applying the log-rank tests. The log-rank tests were used to identify significant differences in the survival functions between the two surgery groups. After checking the proportional hazard assumption, we fitted a univariate and a multivariate Cox proportional hazard model with death as the outcome. All tests were two-tailored, and statistical significance was set at *p* < 0.01. Analyses were carried out using STATA (version 14.0 MP).

## 3. Results

### 3.1. Patients’ Characteristics, Treatment Description, and Perioperative Results

A total of 338 patients (164 males and 174 females) with a mean age of 60.92 years (range: 19 to 94 years) were included in the present study. Among them, 288 patients (85.21%) underwent thoracoscopic pleuropericardial window surgery while 50 patients (14.79%) underwent a subxiphoid pleuropericardial window through mediastinoscopy surgery (see Appendix A, Appendix A for the distribution of all variables by surgery approach).

In terms of dates, for 189 patients (44.08%), the month of diagnosis was missing and we substituted with 1 July, the mid-point of the year. We recorded 18 patients (5.32%) who died in the same month they underwent surgery (results not shown in tables, upon request by authors). The clinical characteristics of patients and diagnosis types by the two types of surgery approaches are presented in Table 1 below.

More patients in the thoracoscopic surgery group (8.68%) were alive at the end of the 2-year survival period than those in the subxiphoid surgery group (6%). There were slightly fewer males (47.92%) than females (52.08%) in the group of thoracoscopic surgery while the distribution of sex among patients in the subxiphoid surgery group was more equal. The mean age of patients in the subxiphoid surgery group was lower (mean = 57 years) than that of patients in the thoracoscopic surgery group (mean = 61 years). Patients who underwent thoracoscopic surgery had been investigated via computed tomography and most patients in the subxiphoid surgery group had been investigated via echocardiography (78%). For patients in the thoracoscopic surgery group, the most frequent clinical diagnosis was pericarditis and left pleural effusion (49.65%), while more than half of patients in the subxiphoid surgery group had pericarditis and right pleural effusion (54%). Hypertension was the most common comorbidity among both groups of patients. Half of the patients in the subxiphoid surgery group had had pericardial tamponade, while this had been seen in only 10 patients (3.47%) in the thoracoscopic surgery group. More than half of the patients in the subxiphoid surgery group had hypodiastolia (80%). Significant differences in imaging diagnosis, presence of pericardial tamponade, and hypodiastolia are observed between thoracoscopic and subxiphoid surgery groups (*p* < 0.001). The patients’ cancer types are presented in Table 2.

Among malignancies, lung cancer was the most encountered, and most patients in both groups of surgery did not present metastases. The pleurodesis treatment and recurrence rate of patients are presented in Table 3.

Talc was used only among patients in the thoracoscopic surgery group, but in a fewer proportion (17.01%) than betaine (82.99%). The recurrence of pleuropericarditis after surgery was experienced only by 10 (3.47%) patients in the thoracoscopic surgery group. Significant differences in pleurodesis treatment by surgery approach are seen (*p* < 0.01). The recurrence of pleuropericarditis after surgery by pleurodesis treatment is shown in Table 4 below.

Pleuropericarditis recurred after surgery only among patients who had received betadine (3.46%). The difference in pleuropericarditis recurrence by pleurodesis treatment is not statistically significant.

In Figure 1, we present the survival probability of the sample by surgery approach, the main independent variable in the study.

We notice that the median survival time for patients in both groups was about 6 months since the date of diagnosis. Slightly more rapid deaths among those who underwent thoracoscopic pleuropericardial window surgery (63%) compared to those who underwent subxiphoid pleuropericardial window surgery (70%) within 3 months since the date of diagnosis are observed; little differences in survival probabilities by surgery approach are noted at later time points. Less than half of the sample, about 40%, were alive at 6 months since the date of diagnosis. Less than a quarter of patients in both groups survived 1 year, which is about 17% of the sample. At 2 years after the date of diagnosis, 8% of those who had undergone thoracoscopic pleuropericardial window surgery survived compared to 6% of those who had undergone subxiphoid pleuropericardial window surgery. However, these differences in the survival curves by surgery type are not statistically significant.

### 3.2. Factors Associated with Mortality

Univariate and Multivariate Cox regression models are shown in Table 5. Model 1 includes age, sex, clinical diagnosis, surgery type, malignancies, and recurrence rate in estimating the survival of patients with pleuropericarditis. Model 2 shows the effect of recurrence alone, and after adjusting for malignancies.

In a univariate Cox proportional hazard regression model, we found that age was associated with reduced survival for patients (Model 1, HR 1.03, 95% CI 1.02–1.03; *p* < 0.001), indicating that for each 1-year increase in age, patients had an increased risk of dying by 3%. In the Cox proportional regression model adjusted for sex, clinical diagnosis, surgery type, and malignancies, age remained statistically significantly associated with the same risk of dying (3%) with each additional year of age (Model 1, HR 1.03, 95% CI 1.02–1.04; *p* < 0.001). In the univariate model, those with leukemia/lymphoma, breast cancer, pleural mesothelioma, and cervical cancer had a significantly lower risk of dying than those with lung cancer. In the adjusted model, the risk of dying among those with leukemia/lymphoma (Model 1, aHR 0.08, 95% CI 0.04–0.13; *p* < 0.001) was 92% lower than those with lung cancer. Those with breast cancer (Model 1, aHR 0.37, 95% CI 0.26–0.52; *p* < 0.001), and those with cervical cancer (Model 1, aHR 0.52, 95% CI 0.27–0.97; *p* = 0.042) had 63%, and 48%, respectively, lower risk of dying than those with lung cancer. Those with pleural mesothelioma (Model 1, aHR 0.32, 95% CI 0.20–0.51; *p* < 0.001) had a 68% lower risk of dying than those with lung cancer. In the univariate model, no difference in the mortality risk was observed by clinical diagnosis, but after adjustment of multivariable factors, the risk of dying among those diagnosed with pericarditis and right pleural effusion (Model 1, aHR 0.76, 95% CI 0.60–0.98; *p* = 0.038) was significantly lower than those diagnosed with pericarditis and left pleural effusion. In the univariate model, patients with pleuropericarditis recurrence had double the risk of dying compared to those who did not experience recurrence recur (Model 1, HR 2.08, 95% CI 1.10–3.94; *p* = 0.023). However, the difference in mortality disappeared once malignancies had been accounted for (Model 2: aHR 1.42, 95% CI 0.75–2.70, *p* = 0.278), underlying the importance of cancer types in explaining the survival of these patients.

## 4. Discussion

Pleuropericarditis is associated with many malignant afflictions, and it is of utmost importance to study the survival of these patients. This study compared the 2-year risk-adjusted survival between two groups of patients with pleuropericarditis: those who underwent thoracoscopic pleuropericardial window surgery and those who underwent a subxiphoid pleuropericardial window surgery through mediastinoscopy. The surgery aimed to efficiently drain both the pleura and pericardium at the same time, prevent the recurrence of the disease, and prolong survival. Along with the drainage, biopsies were taken to establish the type of cancer. This is the first attempt to surgically treat pleuropericarditis in our surgical department. We are not aware of any previous study that surgically treated pleuropericarditis or compared the effect of these two types of surgery approaches on the survival of pleuropericarditis patients. Therefore, the findings of this paper are a starting point for future research to advance the knowledge on post-surgical pleuropericarditis survival.

For 288 patients who presented advanced stages of neoplastic pathologies, with or without tamponade, and without acute cardiac or respiratory failure, we performed a thoracoscopic pleuropericardial window under general anesthesia to allow the neoplastic pericardial effusion to drain in the thoracic cavity [1,6]. Out of these 288 patients, the thoracoscopic pleuropericardial window had a success rate of 96.53%, while 3.47% of the patients (10) had pleuropericarditis recurrence. An 8% recurrence rate of pericardial effusion after thoracoscopy was noted in previous studies that had compared thoracoscopic pericardial window with surgical subxiphoid drainage [19]. However, other studies showed that the pericardial effusion recurrence rate did not differ by surgery type [10]. We cannot compare a recurrence rate of pleuropericarditis with that of pericardial effusion, but we believe that a recurrence proportion of pleuropericardities of about 3.5% is generally low.

As chemical pleurodesis treatment we preferred to use betadine over talc. The literature warns against significant side effects in using chemical pleurodesis with talc, such as fever, chest pain [20], cough, infection at the insertion of the drain, bronchospasm, allergic reaction, arrhythmias, pulmonary edema, pneumonia, respiratory failure, and acute respiratory distress syndrome [21,22]. Other studies found even more disadvantages of talc chemical pleurodesis in malignancies, such as fixation of the lung parenchyma at the level of the pleuropericardial window, which causes pericardial constriction [23,24]. Nonetheless, talc had to be used among patients allergic to betadine.

Only 5% of patients who underwent a thoracoscopic pleuropericardial window had complications, such as seroma, and minimal bleeding. For the 10 patients whose pleuropericardities recurred and for another 40 patients with hypodiastolia (with or without tamponade), the subxiphoid approach under local anesthesia was performed to decompress the pericardial cavity.

This study shows that the survival rate of malignant pleuropericarditis patients who underwent the two types of surgery rapidly decreased after diagnosis. This result may be explained by the fact that in Romania, patients tend to postpone medical check-ups. Most of the time, they wait even years to see a doctor, the visit being driven by pain or severe discomfort. This paper underlines that age, clinical diagnosis, and malignant afflictions are the most important factors related to mortality. Among malignant afflictions, lung cancer was significantly associated with poor survival compared to other types of cancer. Surgery type was not associated with mortality. Our results align with a previous study that showed no significant differences in survival between patients with pericardial effusion who underwent thoracoscopy and those who underwent a subxiphoid pericardial window [10]. However, in Muhammad’s study (2011), patients had only pericardial effusion, the sample size was little, of only 30 patients, and the follow-up after surgery was short (3 months) [10]. Our study has a longer follow-up (two years), a bigger sample size (338 patients), and most importantly, patients had both pericardial and pleural effusions, being the first study to examine post-operative mortality among patients with pleuropericarditis.

This study has several limitations. Firstly, this is a single-center, non-randomized observation study design, and external validation is needed. Moreover, the results of this study cannot be generalized. Secondly, some variables that could be associated with survival or recurrence, such as cancer treatment (chemotherapy, radiotherapy), or the evaluation of concomitant cancer treatments or prior malignancies, were not taken into account. In our thoracic surgery department, we could not collect data about oncological treatment. Moreover, less than half of the patients were alive at 6 months since diagnosis (40%), indicating rapid deaths among the sample, limiting the amount of time to observe and record the evaluation of concomitant cancer treatment, for example. Future studies on post-surgical pleuropericarditis survival may consider these data if available. Finally, unmeasured confounders might have influenced the results of this paper.

## 5. Conclusions

The survival of patients with pleuropericarditis did not differ between patients who underwent a thoracoscopic pleuropericardial window surgery and those who underwent a subxiphoid pleuropericardial window surgery through mediastinoscopy. We performed these two types of surgery approaches to surgically drain both cavities and increase survival. We found that age, having pericarditis and left pleural effusion (compared to having pericarditis and right pleural effusion), and suffering from lung cancer (rather than from leukemia/lymphoma, breast cancer, pleural mesothelioma, and cervical cancer) were the main factors affecting survival in our patients.

## Figures and Tables

**Figure 1 medicina-58-00718-f001:**
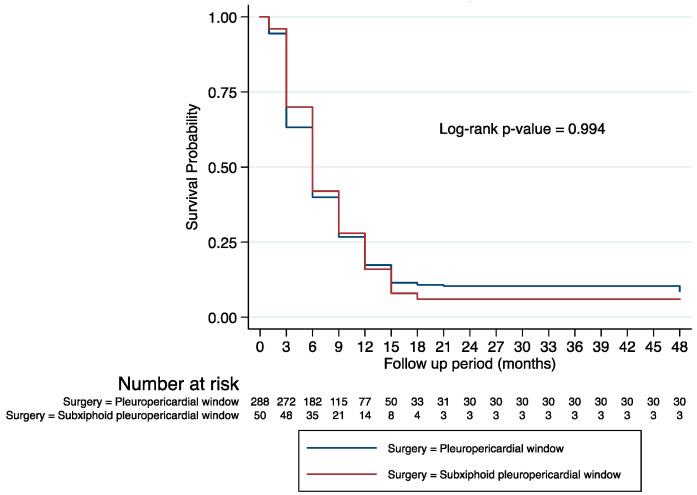
Kaplan–Meier Survival Curves by Surgery Approach.

**Table 1 medicina-58-00718-t001:** The clinical characteristics of patients and diagnosis type by surgery approach.

Variables	Thoracoscopic Pleuropericardial Window (N = 288)	Subxiphoid Pleuropericardial Window through Mediastinoscopy (N = 50)	*p* Value
Event			
Alive/Censored	25 (8.68%)	3 (6.00%)	0.526
Death	263 (91.32%)	47 (94.00%)	
Sex			0.594
Males	138 (47.92%)	26 (52.00%)	
Females	150 (52.08%)	24 (51.48%)	
Age (y)	61 (13.54), 19–94	57 (13.40), 28–80	
Imaging Diagnosis			<0.001
Echocardiography	0 (0.00%)	39 (78.00%)	
Computed tomography	288 (100%)	11 (22.00%)	
Clinical Diagnosis			0.162
Pericarditis and right pleural effusion	114 (39.58%)	27 (54.00%)	
Pericarditis and left pleural effusion	143 (49.65%)	19 (28.00%)	
Pericarditis and bilateral pleural effusion	31 (10.76%)	4 (8.00%)	
Comorbidities			0.449
No comorbidity	57 (19.86%)	14 (28.47%)	
Hypertension	118 (41.11%)	19 (38.78%)	
Diabetes	22 (7.67%)	4 (8,16%)	
Atrial fibrillation	11 (3.83%)	4 (8.16%)	
Renal insufficiency	17 (5.92%)	0 (0.00%)	
Heart failure	48 (16.72%)	6 (12.24%)	
Hepatic insufficiency	7 (2.44%)	1 (2.04%)	
Chronic obstructive bronchopneumpathy	7 (2.44%)	1 (2.04%)	
Presence of pericardial tamponade			<0.001
Yes	10 (3.47%)	25 (50.00%)	
No	278 (96.53%)	25 (50.00%)	
Presence of hypodiastolia			
Yes	0 (0.00%)	40 (80.00%)	<0.001
No	288 (100%)	10 (20.00%)	

Abbreviations: N, number; SD, standard deviation; y, years; N = 338.

**Table 2 medicina-58-00718-t002:** Cancer types of patients by surgery approach.

Variables	Thoracoscopic Pleuropericardial Window (N = 288)	Subxiphoid Pleuropericardial Window through Mediastinoscopy (N = 50)	*p* Value
Malignancies			0.209
Lung Cancer	152 (52.78%)	29 (58.00%)	
Esophageal/Gastric Cancer	5 (1.74%)	1 (2.00%)	
Leukemia/Lymphoma	40 (13.89%)	3 (6.00%)	
Breast Cancer	48 (16.67%)	7 (14.00%)	
Pleural Mesothelioma	19 (6.60%)	2 (4.00%)	
Ovarian Cancer	8 (2.78%)	3 (6.00%)	
Cervical Cancer	10 (3.47%)	1 (2.00%)	
Other types of cancers (Sarcoma, renal and colorectal Cancer)	6 (2.08%)	4 (8.00%)	
Metastases			0.012
No metastases	265 (92.01%)	43 (86.00%)	
Heart	0 (0.00%)	2 (4.00%)	
Liver	6 (2.08%)	0 (0.00%)	
Peritoneum	3 (1.04%)	2 (4.00%)	
Bone	11 (3.82%)	2 (4.00%)	
Upper kidney	2 (0.69%)	1 (2.00%)	
Kidney	1 (0.35%)	0 (0.00%)	

Abbreviations: N, number.

**Table 3 medicina-58-00718-t003:** Pleurodesis treatment and recurrence rates by surgery approach.

Variables	Thoracoscopic Pleuropericardial Window (N = 288)	Subxiphoid Pleuropericardial Window through Mediastinoscopy (N = 50)	*p* Value
Pleurodesis treatment			0.002
Talc	49 (17.01%)	0 (0.00%)	
Betadine	239 (82.99%)	50 (100%)	
Recurrence of pleuropericarditis			0.638
Recurrence	10 (3.47%)	0 (0.00%)	
No recurrence	280 (96.53%)	50 (100%)	

Abbreviations: N, number.

**Table 4 medicina-58-00718-t004:** Recurrence of pleuropericarditis after surgery by pleurodesis treatment.

Pleuropericarditis Recurrence after Surgery	Talc (N = 49)	Betadine (N = 289)	*p* Value
Recurrence	0 (0.00%)	10 (3.46%)	0.186
No recurrence	49 (100%)	279 (96.54%)	

Abbreviations: N, number.

**Table 5 medicina-58-00718-t005:** Cox proportional hazard regression analysis for mortality among patients with pleuropericarditis.

Variable		Univariate HR	95% CI	*p*-Value	Multivariate HR	95% CI	*p* Value
Model 1							
Age	Age	1.03	1.02–1.03	<0.001	1.03	1.02–1.04	<0.001
Sex	Females	1			1		
	Males	1.04	0.83–1.31	0.705	0.99	0.76–1.28	0.930
Clinical diagnosis	Pericarditis and left pleural effusion	1			1		
	Pericarditis and right pleural effusion	0.89	0.71–1.13	0.362	0.76	0.60–0.98	0.038
	Pericarditis and bilateral pleural effusion	1.17	0.80–1.70	0.401	1.01	0.69–1.47	0.975
Surgery	Thoracoscopic pleuropericardial window	1			1		
	Subxiphoid pleuropericardial window through mediastinoscopy	1.00	0.73–1.37	0.996	0.88	0.64–1.23	0.469
Malignancies	Lung Cancer	1			1		
	Esophageal/Gastric Cancer	1.16	0.451–2.62	0.716	1.13	0.49–2.57	0.768
	Leukemia/Lymphoma	0.08	0.04–0.13	<0.001	0.08	0.04–0.13	<0.001
	Breast Cancer	0.43	0.32–0.60	<0.001	0.37	0.26–0.52	<0.001
	Pleural Mesothelioma	0.41	0.26–0.66	<0.001	0.32	0.20–0.52	<0.001
	Ovarian Cancer	0.63	0.34–1.16	0.134	0.79	0.41–1.51	0.485
	Cervical Cancer	0.46	0.25–0.86	0.014	0.52	0.27–0.97	0.042
	Other types of cancer	0.78	0.41–1.48	0.453	0.74	0.39–1.42	0.373
Pleuropericarditis recurrence	No recurrence	1			1		
	Recurrence	2.08	1.10–3.94	0.023	1.22	0.64–2.34	0.534
Model 2							
Pleuropericarditis recurrence	No recurrence	1			1		
	Recurrence	2.08	1.10–3.94	0.023	1.42	0.75–2.70	0.278
Malignancies	Lung Cancer	1			1		
	Esophageal/Gastric Cancer	1.16	0.451–2.62	0.716	1.18	0.52–2.67	0.685
	Leukemia/Lymphoma	0.08	0.04–0.13	<0.001	0.08	0.04–0.13	<0.001
	Breast Cancer	0.43	0.32–0.60	<0.001	0.44	0.32–0.60	<0.001
	Pleural Mesothelioma	0.41	0.26–0.66	<0.001	0.42	0.26–0.66	<0.001
	Ovarian Cancer	0.63	0.34–1.16	0.134	0.63	0.34–1.17	0.149
	Cervical Cancer	0.46	0.25–0.86	0.014	0.46	0.25–0.87	0.016
	Other types of cancer	0.78	0.41–1.48	0.453	0.79	0.41–1.50	0.483

Notes: HR—hazard rate; CI—confidence interval; Model 1 is the full table on which this paper is based; Model 2 shows the effect of pleuropericarditis recurrence on survival, being adjusted only for malignancies.

## Data Availability

The data presented in this study are available on request from the corresponding author.

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
