# Peer review of "The Association of Minimally Invasive Surgical Approaches and Mortality in Patients with Malignant Pleuropericarditis—A 10 Year Retrospective Observational Study"

_medicina, 2022, doi:10.3390/medicina58060718_

Round 1

Reviewer 1 Report

I would like to congratulate the authors for their interesting and informative paper.

This is a single-centre, retrospective observational study investigating the impact of two minimally invasive surgical approaches (i.e., thoracoscopic pleuropericardial window and subxiphoid pleuropericardial window through mediastinoscopy), as well as other factors, on the prognosis of patients with malignant pleuropericarditis. The study included 338 patients from a tertiary acute hospital in Romania between 2009 and 2020. The author found that age, lung cancer, as well as pericarditis with concomitant right sided pleural effusion, were associated with increased risk of mortality. Conversely, surgery type was not associated with mortality risk in this patient population.

Here, I have made a few suggestions that, in my opinion, could help improve the overall quality of the manuscript.

  • The authors may consider describing in more detail how the distinct types of surgical procedures were performed, including relevant technical details for both pericardial and pleural drainage.
  • The authors discuss their preference of povidone iodine over talc for pleurodesis in the Results of the paper. They may consider presenting their arguments in the Discussion section instead.
  • The authors may consider further discussing their findings in reference to more studies investigating malignant pleural or pericardial fluid collections.

Author Response

Here, I have made a few suggestions that, in my opinion, could help improve the overall quality of the manuscript.

  • The authors may consider describing in more detail how the distinct types of surgical procedures were performed, including relevant technical details for both pericardial and pleural drainage.

A detailed description of the two standardized procedures and the management of the postoperative drainage is considered in another paper planned for publication later this year. Due to space limitations, we chose to describe the main details regarding therapeutic management. However, for better clarity, we added in the revised manuscript two footnotes summarizing why we chose one approach over the other (Footnotes 1 and 2, p. 3). Nonetheless, we also list below the criteria for performing these two surgeries:

Selection criteria for the subxiphoid approach under videomediastinoscopic control:

  1. Due to the difficulty with selective orotracheal intubation for anesthesia and the absence of the pleural chamber for thoracoscopic surgery, the subxiphoid approach was easier to perform in approaching the anterior face of the pericardium.
  2. The supine position reduces the pressure created by the pericardial and pleural effusion on the heart and the lung parenchyma.

Patient selection criteria for thoracoscopic approach:

  1. For patients who underwent selective intubation performed by the anesthesiologist, we performed a thoracoscopic approach.
  2. For patients who support the pressure of the pleural and pericardial effusion in the position of lateral decubitus without the appearance of the pathophysiological phenomena of major compression on the mediastinum and of the contralateral lung.
  3. For better visualization of the pleural cavity.

  • The authors discuss their preference for povidone-iodine over talc for pleurodesis in the Results of the paper. They may consider presenting their arguments in the Discussion section instead.

Thank you for this suggestion! We moved the discussion on the reasons for which betadine is preferred over talc in the Discussion section.

  • The authors may consider further discussing their findings in reference to more studies investigating malignant pleural or pericardial fluid collections.

We are not aware of any other studies that compared the survival of patients with pleural or pericardial effusion that had been treated via these two specific surgery approaches. If the reviewer 1 knows about other papers and would like to share those papers with us we would be happy to acknowledge them in our paper.

Reviewer 2 Report

I read the paper with interest and extend my compliments to the authors. The impact of pleuropericarditis on patient with malignant neoplasm is a critical topic to address when choosing a tailored therapy and proper management.

I would have some suggestions for a better interpretation of your paper’s messages and data.

  1. The Abstract: you should preferably avoid the statistical conception and explanation in the abstract, reducing the “materials and methods” part only to clinical or surgical settings and primary or secondary outcomes presentation.
  2. Did you report any preoperative signs of loculated pericardial effusion in your series? If affirmative, did it condition in choosing the surgical approach? 
  3. No intraoperative or postoperative complications rate is mentioned. Is it reliable?
  4. Which Betadine dilution did you use? Did you use a poudrage or a slurry method in the patients where talc had been administered? How many grams of talc did you inflate or inject?
  5. Minor language revision and typos correction are suggested.

Author Response

I read the paper with interest and extend my compliments to the authors. The impact of pleuropericarditis on patient with malignant neoplasm is a critical topic to address when choosing a tailored therapy and proper management.

I would have some suggestions for a better interpretation of your paper’s messages and data.

  1. The Abstract: you should preferably avoid the statistical conception and explanation in the abstract, reducing the “materials and methods” part only to clinical or surgical settings and primary or secondary outcomes presentation.

We thank the reviewer for these suggestions. However, we left the statistical conceptions and their aim in the abstract because we think it is important to state the statistical methods we used to answer to the research aim of the paper. We reduced subsection 2.3 Therapeutic Management in the revised manuscript.

  1. Did you report any preoperative signs of loculated pericardial effusion in your series? If affirmative, did it condition in choosing the surgical approach? 

 We have not only reported signs of neoplasic pericaditis. We reported only signs of pleural and pericardial overflows present simultaneously.

  1. No intraoperative or postoperative complications rate is mentioned. Is it reliable?

We recorded complications only among the group of patients who underwent Thoracoscopic pleuropericardial window; complications noted were seroma, and minimal bleeding. Other major complications were not observed. These complications are mentioned in the Discussion section of the paper.

  1. Which Betadine dilution did you use? Did you use a poudrage or a slurry method in the patients where talc had been administered? How many grams of talc did you inflate or inject?

We used Betadine® (povidone-iodine) in the form of a topical solution 2%, 50 ml (introduced into the pleural cavity at the end of thoracoscopic surgery.

I used talc powder (magnesium silicate), I applied it intraoperatively through poudrage (5 grams of talcum powder) or in the form of a suspension (5 grams of talcum powder in 100 ml of serum) on the thoracic drainage tube. The talcum dust used by us was prepared and sterilized in the hospital pharmacy. I used the medical talcum powder that does not contain asbestos and the particle size was below 50 μm. Sterilization was done by dry heat, gamma rays or ethylene oxide.

  1. Minor language revision and typos correction are suggested.

Thank you for your suggestion; we re-read and corrected the remaining typos.

Reviewer 3 Report

Thank you for the possibility to review the manuscript titled: “The Association of Minimally Invasive Surgical Approaches and Mortality in Patients with Malignant Pleuropericarditis – a 10 Year Retrospective Observational Study”. The study is well written and easy to read. The statistical analysis demonstrates interesting results. There are only a few minor recommendations/corrections:

-Page 3. The term “Gender” should be changed to “sex” as gender is a psychological, not a biological term.

-How did you assess adequate area of resection for the pleuropericardial window?

-Is it possible to assess the side where the pleuropericardial window was performed and include it into the analysis?

Please take into account the recommendations in the spirit of improving the quality of the submission.

Author Response

Thank you for the possibility to review the manuscript titled: “The Association of Minimally Invasive Surgical Approaches and Mortality in Patients with Malignant Pleuropericarditis – a 10 Year Retrospective Observational Study”. The study is well written and easy to read. The statistical analysis demonstrates interesting results. There are only a few minor recommendations/corrections:

-Page 3. The term “Gender” should be changed to “sex” as gender is a psychological, not a biological term.

Thank you for your suggestion. We use now “sex” in the revised manuscript.

-How did you assess adequate area of resection for the pleuropericardial window?

We evaluated the appropriate area for the pleuropericardial window after inspecting the pericardium and pleura. The pericardo-pleural area selected by us was of maximum fluctuation, which allowed us to grip the pericardium so that the heart was not damaged.

-Is it possible to assess the side where the pleuropericardial window was performed and include it into the analysis?

The opening of the pleural cavities by subxiphoid approach through the use of the mediastinoscope was made directed, under video control, with its orientation towards the pleural cavity which had had a pleural effusion. The contralateral pleural cavity, which did not show pleural effusion, was not accidentally opened.

 The detailed description of the two standardized procedures and the management of the postoperative drainage is consider in another paper planned for publication later this year. Due to space limitation, we chose to describe the main details regarding the therapeutic management

Please take into account the recommendations in the spirit of improving the quality of the submission.

We thank reviewer 3 for his/her suggestions!

Reviewer 4 Report

The authors need to be congratulated on their large series of patients with specifically pleural and pericardial malignant effusions as opposed to only pericardial effusion. 338 is indeed a large number. 

The offerings of this paper in my opinion would have been much more attractive and informative had it been presented better. 

There were many things that need to be clarified in the paper. I shall endeavour to name a few.

1) The authors mention one of the operative modalities as "pleuropericardial window through a mediastinoscopy" do they mean mediastinotomy? It would be much better if the authors could describe this operation better. How did they deal with patients with B/L pleural effusions?

2) Are the terms Subxiphoid Pericardial window and mediastinoscopy being used interchangeably? How did the patients who received subxiphoid drainage receive pleural drainage and pleurodesis?

3) Both these procedures being palliative in nature, I wonder if there is any use Querying the differences in survival. This especially when the numbers in the two groups are so vastly different and also there is a clear selection bias where sicker , more unstable patients seem to have received subxiphoid drainage. It would probably be more interesting to look at pleural and pericardial effusion recurrence rates. The authors have briefly touched on that but clarity and in depth analysis is missing.

Author Response

The authors need to be congratulated on their large series of patients with specifically pleural and pericardial malignant effusions as opposed to only pericardial effusion. 338 is indeed a large number. 

The offerings of this paper in my opinion would have been much more attractive and informative had it been presented better. 

There were many things that need to be clarified in the paper. I shall endeavour to name a few.

1) The authors mention one of the operative modalities as "pleuropericardial window through a mediastinoscopy" do they mean mediastinotomy? It would be much better if the authors could describe this operation better. How did they deal with patients with B/L pleural effusions?

The opening of the pleural cavities by subxiphoid approach through the use of the mediastinoscope was made directed, under video control, with its orientation towards the pleural cavity which had had a pleural effusion. The contralateral pleural cavity, which did not show pleural effusion, was not accidentally opened.

 The detailed description of the two standardized procedures and the management of the postoperative drainage is consider in another paper planned for publication later this year. Due to space limitation, we chose to describe the main details regarding the therapeutic management. However, for better clarity we added in the revised manuscript two footnotes summarizing why we chose one approach over the other (Footnote 1 and 2, p. 3). Nonetheless, we also list below the criteria for performing these two surgeries:

Selection criteria for the subxiphoid approach under videomediastinoscopic control:

  1. Due to the difficulty with selective orotracheal intubation for anesthesia and absence of the pleural chamber for thoracoscopic surgery, the subxiphoid approach was easier to perform in approaching the anterior face of the pericardium.
  2. The supine position reduces the pressure created by the pericardial and pleural effusion on the heart and the lung parenchyma.

Patient selection criteria for thoracoscopic approach:

  1. For patients who underwent selective intubation performed by the anesthesiologist we performed thoracoscopic approach.
  2. For patients who support the pressure of the pleural and pericardial effusion in the position of lateral decubitus without the appearance of the pathophysiological phenomena of major compression on the mediastinum and of the contalateral lung.
  3. For a better visualization of the pleural cavity.

2) Are the terms Subxiphoid Pericardial window and mediastinoscopy being used interchangeably? How did the patients who received subxiphoid drainage receive pleural drainage and pleurodesis?

We performed subxiphoid pleuropericardial window through mediastinoscopy surgery. The words „Subxiphoid Pericardial window” and “mediastinoscopy” are not used interchangeably but together to denote one of the two surgical approaches in this paper.

3) Both these procedures being palliative in nature, I wonder if there is any use Querying the differences in survival. This especially when the numbers in the two groups are so vastly different and also there is a clear selection bias where sicker , more unstable patients seem to have received subxiphoid drainage. It would probably be more interesting to look at pleural and pericardial effusion recurrence rates. The authors have briefly touched on that but clarity and in depth analysis is missing.

We thank the reviewer for this observation. We investigated the distribution of pleuropericarditis by surgery approach and we observe that the recurrence of pleuropericarditis after surgery was experienced only by 10 (3.47%) patients in the thoracoscopic surgery group. We undertook deep analysis by adding this variable as a covariate in both the univariate and multivariate the regression models for survival. In the univariate model, patients with pleuropericarditis recurrence had double the risk of dying compared to those who did not experience recurrence recur (Table 4, Model 1, HR 2.08, 95% CI 1.10-3.94; p = 0.023). However, the difference is mortality disappeared once malignancies had been accounted for (Table 4, Model 2: aHR 1.42, 95% CI 0.75-2.70, p = 0.278), underlying the importance of cancer types in explaining the survival of these patients.

Reviewer 5 Report

Thank you for the opportunity to review this retrospective study, which investigates the survival in patients with malignant pleuropericarditis treated with subxiphoid or thoracosopic minimally invasive surgery. The topic is interesting and the manuscript is quite original, as it includes patients with both pleural and pericardial effusion. However, I think the authors could better highlight the novelty of the manuscript by describing their centre’s approach more systematically and by including more data describing the study population. Moreover, it is not clear through the manuscript whether the authors’ aim was to assess the mortality of patients with malignant pleuropericarditis, or to compare the two surgical techniques in terms of survival (in this case, results should be presented separately for the two groups in order to understand the differences).

Here are some comments:

Introduction

-“This study is a first attempt to investigate the risk factors of patients with pleuropericarditis who underwent either minimally invasive thoracoscopic pleuropericardial window or subxiphoid pleuroperi-cardial window through mediastinoscopy surgery.”: the authors should be more specific, are they investigating risk factor for mortality?

Materials and methods

-I think the methods section could be better organized. In fact, I believe the authors should not include any result in this paragraph (i.e. the number of patient diagnosed with malignant pleuropericarditis during the study period), nor describe patients’ presentation at the time of admission. Data should always be included in the results section.

-Among the inclusion criteria, when the authors say “pathology report”, do they mean the cytology was always performed on the pericardial fluid to confirm the malignant nature of the effusion before surgery?

-“The pleural and pericardial effusions were found in an advanced stage of the disease in all patients”: I think the meaning of this sentence is not clear.

-“we decided to use the subxiphoid minimally invasive approach for patients with hypo-diastolia”: the author discuss this decision as this was taken for a study protocol: however, this is a retrospective study. I believe it would be better if the authors use the expression “at our centre, the subxiphoid approach is employed in case of….” or something similar. Moreover, where the left and right pleurae always opened during subxiphoid mediastinoscopy or was it only done in case of bilateral pleural effusion? How was the thoracic drainage managed? Maybe the authors could describe the two standardized procedures and post-operative drainage management, and describe the objective criteria for using one approach or the other.

-Page 3 “Among these 35 patients with pericardial tamponade, 28.57% (10) underwent thoracoscopic pleuropericardial window while 71.42% (25) underwent subxiphoid pleuropericardial window surgery through mediastinoscopy.”: the sum does not add up to 100%.

-Page 3 “We used betadine in 289 cases (85.50%) and talc in 49 cases (14.50%). Betadine was chosen because of its higher efficiency compared to that of talc (96% vs 73%) [16, 17, 18].”: I did not find reference 16 to report a higher efficacy of betadine compared to talc; reference 17 does not seem compare the two procedures; I was not able to find the full text of reference 18, but the abstract does not mention the use of betadine. Could the authors please indicate where they found this difference in terms of efficiency between betadine and talc?

-“Survival was defined as the time from the date of diagnosis to date-of-death.”: do the authors mean the diagnosis of pleuropericardial effusion or the diagnosis of cancer? For how many patients the month was not indicated (thus impyling that the survival could be right or differ up to 6 months)?

-The authors should clearly state the main (and secondary aims) of the study: is it to compare survival between the thoracoscopic and subxiphoid surgery groups? Is it to assess survival in patients with surgically-managed pleuropericarditis?

Results

-I understand that this is a retrospective study and that the authors could not collect data on oncological treatment after surgery , but there are very few data describing the study population: could the author please include some data, for example with regard to comorbidities, symptoms at presentation, echocardiographic and CT findings, any previous chemo-radiotherapy/thoracic surgery, patients’ performance status at the time of diagnosis, neoplastic stage (for example patients with metastasis in sites other than the pleura/pericardium), time from the diagnosis of cancer to the development of pleuropericarditis?

-The author should clarify whether “recurrence of pleuropericarditis” indicates patients which experienced a new episode after treatment or patients who had already received treatment for pleuropericarditis before the period of the study. I believe type of patients should be described in the paper. The author say 2,96% of patients experienced recurrence, how were they distribute in the two groups?

- “This suggests that patients with pleuropericarditis experienced rapid deaths since being diagnosed. “ I think considerations and comments on the results should only be included in the “discussion” paragraph

-Were there differences in terms of recurrence between patients receiving pleurodesis with betadine or talc?

Discussion

- “Moreover, to the best of our knowledge, our surgical management of pleurocardities represents a new diagnostic and treatment approach.”: I think the authors should thoroughly describe their management in order to support and highlight the novelty of this study.

- “This study shows that survival rate among malignant pleuropericarditis was drastically reduced after diagnosis”: I do not understand this sentence, the authors say that survival is “reduced” but compare to what?

-Could the authors please include post-operative complications and their severity in the results?

-“ An 8% recurrent rate of pericardial effusion after thoracoscopy was noted in previous studies which compared thoracoscopic pericardial window with surgical subxiphoid drainage [24].”: could the author compare their recurrent rate with that reported in the literature?

Conclusions

-Could the authors please start the conclusions by answering to the primary aim of the study?

-Table 1: data such as clinical diagnosis, surgery type,… are not part of the patient demographics. Could the authors please fix this? please also check the sum of the percentages, it does not always add up to 100%. Moreover, if the aim of the study is to compare survival between two groups, I believe data should be divided for the two groups, so as to describe the population and the homogeneity.

-Figure 2: maybe the contents of the 5 part of the figure (A,B,C,D,E) should be separately described in the figure caption/legend

Author Response

Comments and Suggestions for Authors

Thank you for the opportunity to review this retrospective study, which investigates the survival in patients with malignant pleuropericarditis treated with subxiphoid or thoracosopic minimally invasive surgery. The topic is interesting and the manuscript is quite original, as it includes patients with both pleural and pericardial effusion. However, I think the authors could better highlight the novelty of the manuscript by describing their centre’s approach more systematically and by including more data describing the study population. Moreover, it is not clear through the manuscript whether the authors’ aim was to assess the mortality of patients with malignant pleuropericarditis, or to compare the two surgical techniques in terms of survival (in this case, results should be presented separately for the two groups in order to understand the differences).

Here are some comments:

Introduction

-“This study is a first attempt to investigate the risk factors of patients with pleuropericarditis who underwent either minimally invasive thoracoscopic pleuropericardial window or subxiphoid pleuroperi-cardial window through mediastinoscopy surgery.”: the authors should be more specific, are they investigating risk factor for mortality?

We have changed the last two sentences at the end of the introduction into “To the best of our knowledge, this study is a first attempt to treat malignant pleuropericarditis with minimally invasive thoracoscopic pleuropericardial window and subxiphoid pleuropericardial window through mediastinoscopy surgery, and to compare the survival of these patients by the two types of surgery approaches.”.

We would like to clarify that, in this article, the authors do not investigate the risk factors for pleuropericardial effusion or for mortality among pleuropericardial effusion.

Also, the chosen surgical technique does not influence the risk factors of neoplastic disease and mortality.

The goal of minimally invasive surgery was to reduce heart and respiratory failure, reduce postoperative patient suffering, and reduce the recurrence of pleuropericarditis.

Materials and methods

-I think the methods section could be better organized. In fact, I believe the authors should not include any result in this paragraph (i.e. the number of patient diagnosed with malignant pleuropericarditis during the study period), nor describe patients’ presentation at the time of admission. Data should always be included in the results section.

Thank you for your suggestion! We took out the numbers from the section describing the therapeutic management. More data describing patients’ characteristics has been added in the Results section

-Among the inclusion criteria, when the authors say “pathology report”, do they mean the cytology was always performed on the pericardial fluid to confirm the malignant nature of the effusion before surgery?

The diagnosis of malignancy in malignant pleuropericarditis was established by taking biopsies collected by thoracoscopy or by mediastinoscopic control in the subxiphoid approach of the pleura and pericardium. These biopsies were examined anatomopathologically.

-“The pleural and pericardial effusions were found in an advanced stage of the disease in all patients”: I think the meaning of this sentence is not clear.

We deleted this sentence but we meant that the concomitant presence of pleural and pericardial effusions is an advanced stage of the underlying neoplastic disease.

-“we decided to use the subxiphoid minimally invasive approach for patients with hypo-diastolia”: the author discuss this decision as this was taken for a study protocol: however, this is a retrospective study. I believe it would be better if the authors use the expression “at our centre, the subxiphoid approach is employed in case of….” or something similar. Moreover, where the left and right pleurae always opened during subxiphoid mediastinoscopy or was it only done in case of bilateral pleural effusion? How was the thoracic drainage managed? Maybe the authors could describe the two standardized procedures and post-operative drainage management, and describe the objective criteria for using one approach or the other.

We changed the sentence into “At our center, the subxiphoid minimally invasive approach is employed for patients with hypodiastolia.”

The opening of the pleural cavities by subxiphoid approach through the use of the mediastinoscope was made directed, under video control, with its orientation towards the pleural cavity which had had a pleural effusion. The contralateral pleural cavity, which did not show pleural effusion, was not accidentally opened.

 The detailed description of the two standardized procedures and the management of the postoperative drainage is consider in another paper planned for publication later this year. Due to space limitation, we chose to describe the main details regarding the therapeutic management. However, for better clarity we added in the revised manuscript two footnotes summarizing why we chose one approach over the other (Footnote 1 and 2, p. 3). Nonetheless, we also list below the criteria for performing these two surgeries:

Selection criteria for the subxiphoid approach under videomediastinoscopic control:

  1. Due to the difficulty with selective orotracheal intubation for anesthesia and absence of the pleural chamber for thoracoscopic surgery, the subxiphoid approach was easier to perform in approaching the anterior face of the pericardium.
  2. The supine position reduces the pressure created by the pericardial and pleural effusion on the heart and the lung parenchyma.

Patient selection criteria for thoracoscopic approach:

  1. For patients who underwent selective intubation performed by the anesthesiologist we performed thoracoscopic approach.
  2. For patients who support the pressure of the pleural and pericardial effusion in the position of lateral decubitus without the appearance of the pathophysiological phenomena of major compression on the mediastinum and of the contralateral lung.
  3. For a better visualization of the pleural cavity.

-Page 3 “Among these 35 patients with pericardial tamponade, 28.57% (10) underwent thoracoscopic pleuropericardial window while 71.42% (25) underwent subxiphoid pleuropericardial window surgery through mediastinoscopy.”: the sum does not add up to 100%.

Thank you for your observation, we have corrected into 71.43%.

-Page 3 “We used betadine in 289 cases (85.50%) and talc in 49 cases (14.50%). Betadine was chosen because of its higher efficiency compared to that of talc (96% vs 73%) [16, 17, 18].”: I did not find reference 16 to report a higher efficacy of betadine compared to talc; reference 17 does not seem compare the two procedures; I was not able to find the full text of reference 18, but the abstract does not mention the use of betadine. Could the authors please indicate where they found this difference in terms of efficiency between betadine and talc?

We clarified with the following sentence: “Since betadine and talc are among the most used chemical agents for chemical pleurodesis [16, 17, 18], we used betadine in most patients (see Table 5) and talc among those allergic to betadine.” (p.3) We moved the discussion on the reasons for which betadine is preferred over talc in the Discussion section.

-“Survival was defined as the time from the date of diagnosis to date-of-death.”: do the authors mean the diagnosis of pleuropericardial effusion or the diagnosis of cancer? For how many patients the month was not indicated (thus impyling that the survival could be right or differ up to 6 months)?

In the Measurements section, we clarified that the survival was defined as the time from the date of diagnosis of pleuropericardial effusion.

In the Results section, we also clarified that among 338 patients, for 189 (44.08%) we did not find the month of the diagnosis of pleuropericarditis and we substituted the missing values for the month with the date of 1st July, representing the mid-year. We added in the Results section that 18 patients died in the same month as the clinical diagnosis (they contribute with 1 month in the survival analysis). We want to reassure the reviewer that this approach is well encountered in survival analysis (Alison, 1984).

-The authors should clearly state the main (and secondary aims) of the study: is it to compare survival between the thoracoscopic and subxiphoid surgery groups? Is it to assess survival in patients with surgically-managed pleuropericarditis?

We thank the reviewer for this observation. The main aim of the study is indeed to compare survival among patients with cancer who underwent the thoracoscopic and subxiphoid surgeries. We have clarified this throughout of the paper.

Results

-I understand that this is a retrospective study and that the authors could not collect data on oncological treatment after surgery , but there are very few data describing the study population: could the author please include some data, for example with regard to comorbidities, symptoms at presentation, echocardiographic and CT findings, any previous chemo-radiotherapy/thoracic surgery, patients’ performance status at the time of diagnosis, neoplastic stage (for example patients with metastasis in sites other than the pleura/pericardium), time from the diagnosis of cancer to the development of pleuropericarditis?

We added data for imaging diagnosis, metastases, presence for pericardial tamponade, and for hypodiastolia. Unfortunately, we do not have other data suggested by the reviewer. Nonetheless, we acknowledged in the limitations of the study that variables such as cancer treatment, evaluation of concomitant cancer treatments or prior malignancies may affect the results.

-The author should clarify whether “recurrence of pleuropericarditis” indicates patients which experienced a new episode after treatment or patients who had already received treatment for pleuropericarditis before the period of the study. I believe type of patients should be described in the paper. The author say 2,96% of patients experienced recurrence, how were they distribute in the two groups?

We clarified in the Measurements section that we refer at Recurrence of pleuropericarditis after surgery. We had already stated in the section Therapeutic Management that recurrence of pleuropericaditis occurred in 10 patients who underwent thoracoscopic pleuropericardial window surgery. We are now showing the results section how the recurrence of pleuropericarditis after surgery differ by surgery type.

- “This suggests that patients with pleuropericarditis experienced rapid deaths since being diagnosed. “ I think considerations and comments on the results should only be included in the “discussion” paragraph.

This sentence is now deleted from the Results section.

-Were there differences in terms of recurrence between patients receiving pleurodesis with betadine or talc?

Table 1: Recurrence of pleuropericarditis after surgery by pleurodesis treatment

Recurrence of pleuropericarditis after surgery

Talc

Betadine

Total

Recurrence

0 (0%)

10 (3.46%)

10 (2.96%)

No recurrence

49 (100%)

279 (96.54%)

328 (97.04%)

Total

49 (100%)

289 (100%)

338 (100%)

We notice that more patients experience pleuropericarditis recurrence among those who had been treated with betadine (96.54%) compared to those who had been treated with betadine (3.46%).

We did not add this detail/table in the paper.

Discussion

- “Moreover, to the best of our knowledge, our surgical management of pleurocardities represents a new diagnostic and treatment approach.”: I think the authors should thoroughly describe their management in order to support and highlight the novelty of this study.

We changed the sentence into “This is the first attempt to surgically treat pleuropericarditis in our surgical department.”

We added more variables in the paper, presented in the Results section by surgery approach, describing patients’ characteristics, diagnosis types, patients’ cancer types, pleurodesis treatment, and recurrence rate.

- “This study shows that survival rate among malignant pleuropericarditis was drastically reduced after diagnosis”: I do not understand this sentence, the authors say that survival is “reduced” but compare to what?

We changed this sentence with “This study shows that survival rate among malignant pleuropericarditis rapidly decreased after diagnosis” (p. 9). We want to underline how fast survival rates decreased after the surgery.

-Could the authors please include post-operative complications and their severity in the results?

We recorded complications only among the group of patients who underwent VATS pleuropericardial window; complications noted were seroma, and minimal bleeding. Other major complications were not observed. These complications are mentioned in the Discussion section of the paper.

-“ An 8% recurrent rate of pericardial effusion after thoracoscopy was noted in previous studies which compared thoracoscopic pericardial window with surgical subxiphoid drainage [24].”: could the author compare their recurrent rate with that reported in the literature?

We cannot compare pleuropericarditis with pericardial effusion, so we cannot compare the two recurrence rates. However, we wrote the following sentence “We cannot compare a recurrence rate of pleuropericarditis with that of pericardial effusion, but we believe that a recurrence proportion of pleuropericardities of about 3.5% is generally low.”.

Conclusions

-Could the authors please start the conclusions by answering to the primary aim of the study?

Thank you, we now start the conclusion with “The survival of patients with pleuropericarditis did not differ between patients who underwent thoracoscopic pleuropericardial window surgery and those who underwent subxiphoid pleuropericardial window surgery through mediastinoscopy.”

-Table 1: data such as clinical diagnosis, surgery type,… are not part of the patient demographics. Could the authors please fix this? please also check the sum of the percentages, it does not always add up to 100%. Moreover, if the aim of the study is to compare survival between two groups, I believe data should be divided for the two groups, so as to describe the population and the homogeneity.

Thank you for this observation; we check all numbers and we made sure they add up to 100%. We also included more variables that we grouped into

  1. a) patients characteristics and diagnosis types
  2. b) patients’ cancer types
  3. c) pleurodesis treatment and recurrence rate

-Figure 2: maybe the contents of the 5 part of the figure (A,B,C,D,E) should be separately described in the figure caption/legend

We decided to keep only one figure in the paper, describing the survival curves by surgery type, the main focus on the paper.

Round 2

Reviewer 4 Report

I do not feel that the questions raised in my previous review have been adequately addressed. For example, the authors still have not described their operative techniques. I am still yet to understand how the authors drained pleural effusion (right or left via a mediastinoscopy?) or do the authors mean mediastinotomy?

Author Response

See document attached

Reviewer 5 Report

I want to thank the authors for making all these changes to the manuscript. I have some comments:

-when indicating data in the text one should always indicate the unit of measurement (i.e. “mean =57” page 5). Also, when reporting a difference between the two groups, it would be better if you highlighted whether the difference reached statistical significance.

- Page 5 “For patients in the thoracoscopic surgery group, the most frequent clinical diagnosis was pericarditis and left pleural effusion (49.65%) while for those in the other group it was pericarditis and left pleural effusion (57%).” I believe there is an error here, 54% of patients in the subxiphoid-approach group had pericarditis and right pleural effusion (not left and not 57%).

-I thank the authors for including more information, however I am still concerned by the lack of data on oncological treatment before and after surgery, performance status etc. which have a major impact on the significance of this study.

-I believe the fact that every patient experiencing recurrence after treatment had received pleurodesis with betadine it’s interesting and should be mentioned in the study.

Author Response

See document attached
